# Corollary discharge and efference copy mechanisms in schizophrenia and controls: The N1 and P2 evoked potential components differentially react to self-initiated tones in schizophrenia

Constantin Liermann-Koch [1,2,3]*, Jan W. Thielen[1,2], Dae-In Chang[4], Richard Krieger-Strásky[5], Oliver Kraff[6], Norbert Scherbaum[1,2], Emma Sprooten[7,8], Bernhard W. Müller [1,2,9]

1 LVR-Hospital Essen, Department of Psychiatry and Psychotherapy, Faculty of Medicine, University of Duisburg-Essen, Essen, Germany, 2 Center for Translational Neuro- and Behavioral Sciences (C-TNBS), Medical Faculty, University Hospital of Duisburg-Essen, Essen, Germany, 3 Department of Cardiology, Internal Intensive Care Medicine & Emergency Medicine, Klinikum Dortmund, Dortmund, Germany, 4 LWL-Hospital University, Department of Psychiatry, Psychotherapy and Preventive Medicine, Medical Faculty of the Ruhr University Bochum, Bochum, Germany, 5 Psychedelic Research Centre, National Institute of Mental Health, Klecany, Czech Republic, 6 Erwin L. Hahn Institute for Magnetic Resonance Imaging, University of Duisburg-Essen, Essen, Germany, 7 Department of Human Genetics, Radboud University Medical Center, Nijmegen, The Netherlands, 8 Department of Cognitive Neuroscience, Donders Institute for Brain, Cognition and Behavior, Radboud University Nijmegen Medical Center, Nijmegen, The Netherlands, 9 Department of Psychology, University of Wuppertal, Wuppertal, Germany

* Constantin.liermann-koch@stud.uni-due.de

## Abstract

"Efference copy" and "corollary discharge" mechanisms may help to distinguish internal from external stimuli. Previous studies indicate that patients with schizophrenia show a lack of N1 evoked response potential component amplitude suppression to self-initiated auditory stimuli, suggesting a corollary discharge impairment. In our study, we examined N1 and P2 component amplitude suppression in 27 patients with schizophrenia and 30 healthy controls using an auditory button-press paradigm. In addition to symptom ratings we conducted neuropsychological assessments. Our findings replicated N1 amplitude suppression to self-generated tones in schizophrenia patients. We found no correlation between N1, P2, and lateralized readiness potential, suggesting that the readiness potential may not directly represent corollary discharge mechanisms. Cognition performance was reduced in schizophrenia patients and performance reduction correlated with negative symptoms. Cognition was not associated with evoked potential data. Regarding the P2 component, P2 suppression to self-generated tones was observed in patients as well as in controls. In conclusion, the N1 and the P2 component seem to be differentially involved in corollary discharge and efference copy mechanisms. Further investigation is needed to elucidate functional differences and sources of both components in this context.

**Data availability statement:** The EEG raw dataset, EEG analysis dataset, as well as the PANSS, WAIS and CCAS datasets, are available from the DuEPublico database via https://doi.org/10.17185/duepublico/84480.

**Funding:** This study was partly funded by a NARSAD Young Investigator Award of the Brain and Behavior Research Foundation (BBRF), 90 Park Avenue, 16 th floor, New York, NY, USA. for P.I. Emma Sprooten, Ph.D; Grant ID: 25034. The publication fee was funded by the Open Access Publication Fund of the University of Duisburg-Essen. The funders had no role in study design, data collection and analysis, decision to publish, or preparation of the manuscript.

**Competing interests:** The authors have declared that no competing interests exist.

## Introduction

Schizophrenia is a neuropsychiatric disorder with an estimated annual incidence rate of approximately 0.01% with onsets, occurring most frequently in young adults between the ages of 20 and 35 [1,2]. This condition leads to elevated socioeconomic expenses, diminished quality of life, and a lower life expectancy [3,4]. Schizophrenia is characterized by positive or negative symptoms (e.g., hallucinations, and lack of motivation), motor abnormalities, and cognitive and social deficits. Although the etiology of schizophrenia is incompletely understood, part of the symptoms may be explained by the corollary discharge/efference copy hypothesis proposed by Feinberg [5]. It is hypothesized that hallucinations may be caused by a disrupted internal forward model that impairs one's ability to distinguish between internally and externally generated events. At the same time, the inability to predict self-generated sensations has been suggested to be related to the experience of auditory verbal hallucinations [6].

More specifically, according to Frith's model [6], the efference copy (EC) informs the sensory area of the planned movement, where the actual event is then compared to the planned movement corollary discharge (CD). In this context, it has been suggested that the process may be disrupted in individuals with schizophrenia [7,8]. The theory of EC/CD dysfunction in schizophrenia was repeatedly tested in experiments using EEG-derived event-related potentials. In a first study, Ford et al. [9] examined the effects of speech on the N1 auditory component and found that the amplitude of the N1 component was not suppressed in the schizophrenia patients, unlike in the control subjects. The N100 or N1 component of the auditory-evoked potential is a large, negative-going evoked potential measured by electroencephalography; it peaks between 80 and 120 ms after stimulus onset, and its amplitude is known to be suppressed in response to self-generated auditory stimuli [10,11], but not in patients with schizophrenia [9]. Therefore, a lack of N1 attenuation indicates potential self-monitoring deficits [12] that may lead to misattribution of an internal self-generated source to an external source. The same effect was subsequently observed across various motor and sensory modalities [13–18]. Additionally, these neuroscientific underpinnings were reexamined and developed in the context of animal studies [19–22].

As described above, EC information regarding planned movements relates to future action-motor planning. These processes have been evaluated through *Bereitschaftspotential* or readiness potential (RP) research, initially described by Kornhuber and Deecke in 1965 [23]. RP is a slow, negative cortical potential that develops 2,000 ms to 1,000 ms prior to a self-initiated movement [24] – Shibasaki's 2006 paper [25] provides further details. In patients with schizophrenia, studies have reported that RP amplitudes are abnormal and reduced when compared with those of healthy controls [26–31].

Both evoked potential components, N1 and RP, were first investigated by Ford et al. [32], who assessed three conditions: motor + auditory, auditory-only, and motor-only. In the motor + auditory condition, an auditory stimulus was administered with each button press; in the auditory-only condition, an external sequence of tones was presented to the participant while they remained stationary; and in the motor-only condition, the participant pressed a button, but no sounds were presented. To enable

a comparison between the motor + auditory and auditory-only conditions, the motor-only condition was subtracted from the motor + auditory condition to control for motion-related effects. The results indicated that patients with schizophrenia exhibited less suppression of N1 when performing a motor action that produced a sound. Additionally, the study revealed a correlation between deficits in the suppression of the auditory N1 component and impairment of the lateralized readiness potential (LRP) prior to a self-paced button press when all subjects were included and not separated by group.

While the mechanisms of EC/CD and their disruption in schizophrenia have been extensively studied, further publications followed, particularly in the area of deep learning [33], with the publication of the data from the Ford et al. 2014 study by one of the co-authors [34]. To our knowledge, there is no comparable study in an independent patient population, although similar studies exist in terms of the procedures employed [35,36], but with a different focus, namely the additional variable time-shifted playback of tones after key presses, a different study population and without measurement of the readiness potential.

Thus, we aimed to replicate the Ford 2014 study [32]. Our first objective was to assess whether self-initiated auditory events lead to a lack of N1 suppression in patients with schizophrenia. The second objective was to determine if the readiness potential (RP) is reduced in these patients and its relationship with N1 suppression. Lastly, we conducted a brief neuropsychological assessment to explore its associations with EC/CD in schizophrenia and healthy controls. We predicted: (1) replication of N1 suppression for self-reported tones, (2) reduced RP associated with N1 suppression in patients, and (3) a correlation between ERP changes and cognitive test results [37].

## Method

This study was conducted at the Department of Psychiatry and Psychotherapy, University of Duisburg-Essen, Germany. It was approved by the Ethics Committee of the Medical Faculty of the University of Duisburg-Essen (No. 01-17-7753-B0). Recruitment took place from 1st February 2019–30th April 2023. All participants provided their written informed consent before participating. A trained research assistant, psychiatrist, or clinical psychologist administered all assessments, including the EEG, PANSS, WAIS-IV, and CCAS, within three weeks.

In our study we included 28 adult patients with ICD-10 schizophrenia and 30 healthy controls. Diagnoses were based on the structured clinical interview for ICD-10 and have the following subtypes: 14 diagnoses of paranoid schizophrenia, seven of undifferentiated schizophrenia, three of residual schizophrenia, three of schizoaffective disorder, and one of hebephrenic schizophrenia. Patients with schizophrenia (SG) were recruited from the schizophrenia inpatient facility at the University of Duisburg-Essen's Department of Psychiatry and Psychotherapy, Germany. At the time of testing, patients had to be clinically stable (not in acute exacerbation) and should not have had a change in medication in the preceding four weeks. Healthy controls (CG) were recruited through advertisements and word of mouth. Data from individuals with schizophrenia were analyzed alongside data from 30 healthy controls matched for age and gender. One SG ERP data point was removed from the analysis due to technical issues. Both groups received monetary compensation for their participation. The inclusion criteria for both groups were as follows: have an intelligence quotient greater than 75 (as measured by two subtests of the Wechsler Abbreviated Scale for Intelligence) and be older than 18 years but younger than 45 years. For the patient group, additional inclusion criteria are diagnoses based on the structured clinical interview for ICD-10. Exclusion criteria for both groups were a known history of neurological or medical disease, traumatic brain injury, history of substance dependence, current (same day) abuse, or intoxication of substances other than nicotine. All patients were treated according to the German Mental Health Act ("Psychisch-Kranken-Gesetz").

### Positive and negative syndrome scale (PANSS)

The 30-item PANSS [38] comprises four scales measuring positive and negative syndromes, their differential, and the general severity of illness. The interviewers administered the PANSS as part of a structured clinical interview and rated items on a scale from 1 (asymptomatic) to 7 (extremely symptomatic).

### Cerebellar cognitive affective/Schmahmann syndrome (CCAS) scale

This test consists of 10 sub-tests and provides a total raw score, cut-offs for each test, and pass/fail criteria that determined "possible" (one test failed), "probable" (two tests failed), and "definite" (three tests failed) CCAS [39]. We used the German version [40]. This test is a compilation of several established, well-known tests. These are the Semantic Test, Phonemic Test, Category Switching, Verbal Registration/Verbal Recall, Digit Span Forward, Digit Span Backward, drawing a Cube, Similarities, Go-No-Go, and Affect.

### WAIS-IV/ Wechsler adult intelligence scale – Fourth edition

WAIS-IV includes 15 subtests across four index scales. Our study evaluated the Verbal Comprehension Index, which focuses on vocabulary, and the Perceptual Reasoning Index, which is based on matrix reasoning [41]. We utilized the German version [42].

### Apparatus

We used BrainVision Recorder software (version 1.1, Brain Products, GmbH, Germany) to continuously record data at a rate of 1,000 Hz within a frequency band ranging from 0.016 Hz to 250 Hz. Active shielded electrodes (ActiCap Slim Ag/AgCl), which provide extra shielding at the electrode, were connected to a Brain Products BrainAmp DC amplifier. Electrodes were positioned on the scalp using an elastic cap (EasyCap) based on the international 10–20 system and placed at the following positions: Fp1, Fp2, F7, F3, Fz, F4, F8, FC1, FC2, FC5, FCz, FC6, T7, C3, Cz, C4, T8, CP5, CP1, CP2, CP6, P7, P3, Pz, P4, P8, O1, OZ, and O2. Additional electrodes were placed on the left and right earlobes, with FCz serving as the reference electrode and Fpz as the ground electrode. Horizontal eye movements were recorded using two electrodes at the outer corners of the left and right eyes, for the purpose of electroculography artifact correction. With this setup, impedances were kept below 25 kilohms.

### Procedure/Task

In a quiet and electrically shielded room, participants were seated in a comfortable recliner. After the electrode cap and EEG/EOG electrodes were attached, each recording session generally lasted approximately 50 min, during which three experimental conditions were completed. In the first condition, "Generate Tone," participants were asked to press a button with their right index finger every five to seven seconds to produce a short 80-ms, 1,000-Hz tone with a 10-ms rise and fall time using Telephonics TDH-39P headphones. The tone had a sound pressure level of 80 dB, and there was zero delay between the button press and the onset of the tone. The time parameters of button presses were recorded for use in the second condition. In the second condition ("Hear Tone"), participants listened to their own temporal sequence of tones from the first condition ("Generate Tone") without finger movements. The third condition was identical to the first condition except that participants did not produce an acoustic response when pressing the button ("Button Alone"). This condition is similar to the RP paradigm [23]. Prior to each session, participants completed a training block to familiarize themselves with the tasks. Moreover, they were asked to avoid excessive eye and head movements. Each condition was completed after a minimum of 100 trials.

### Data acquisition, preprocessing, and statistical analysis

We used Brain Vision Analyzer, Brain Products GmbH, Munich, Germany (version 2.2) for data analysis. After the EEG was re-referenced to connected earlobes, we performed visual inspection to reject data contaminated by gross artifacts. Data were digitally filtered using a finite impulse response filter between 0.027 Hz and 30 Hz with an order of eight. Eye movement artifacts were corrected using independent component analysis [43]. Continuous EEG data were segmented into 3,000-ms epochs (−2,000ms to +1,000ms) time-locked to button presses (coincident with tone onset). EEG epochs

were each artifact rejected for voltages greater than ±120 μV and then averaged. For the N1 and P2 analyses, the data were baseline corrected from −100–0 ms. N1 and P2 peaks were identified at electrodes Fz, FCz, and Cz. To account for the motion-related effects of the "Generate Tone" condition in the statistical analyses, we removed the "Button Alone" condition from the "Generate Tone" condition. For the analysis of RP in the "Generate Tone" and "Button Alone" conditions, the data were baseline corrected from −2,000 to −1,800 ms and subsequently divided into ten 200-ms segments from −2,000 ms to 0 ms to derive the average amplitude values.

Statistical analysis was performed using SPSS software (version 29). We employed Student's *t*-test to analyze the demographic, button-pressing pace, average number of included trials, and neuropsychological tests. N1 and P2 were analyzed in a three-way repeated measures ANOVA for group ("CG" vs "SG") as a between-subject factor and condition ("Generate Tone" versus "Hear Tone") and electrode (Fz, FCz, and Cz) as within-subject variables. Furthermore, RPs were analyzed in a three-way repeated measures ANOVA with group ("CG" vs "SG") as between-subjects factor and conditions ("Generate Tone" versus "Button Alone"), electrodes (F3, Fz, F4, C3, Cz, C4, and FCz) and the 10 segments (−2,000 ms to 0 ms in 200 ms intervals) as within-subject variables. The correlations "ERP Measures versus LRP Amplitude", "Relationship between Clinical Symptoms and ERP Measures," "Relationship between PANSS and Cognitive Performance" and "Relationship between Illness Duration and cognitive performance" were calculated using Pearson's correlation coefficient.

## Results

### Demographics, WAIS-IV, and CCAS score

The demographic and clinical characteristics of participants are listed in Table 1. There were no significant differences between the SG and CG groups in terms of age, sex ratio, but years of education ($M_{Diff}$: −2.11, 95%-CI [- 4.09, −0.123], $t(55) = 0.038$, $p = 0.038$).

The CCAS results are presented in Table 2. On the CCAS scale, patients with schizophrenia showed a statistically significant lower total score compared with the control group ($M_{Diff}$: −18.4, 95% CI [−26.79, −10.16], $t(55) = −4.45$, $p < 0.001$. In addition, the CCAS scale revealed a significantly higher total number of failed tests among patients with schizophrenia than in the control group (2.16, 95% CI [1.31, 3.02], $t(55) = 5.08$, $p < 0.001$).

On the WAIS-IV, the raw matrix reasoning score was significantly lower among patients with schizophrenia than in the control group ($M_{Diff}$: −3.03, 95% CI [−5.16, −0.90], $t(55) = −2.85$, $p = 0.006$). Similarly, the raw vocabulary score was significantly lower in the control group ($M_{Diff}$: −8.58, 95% CI [−13.85, −3.30], $t(55) = −3.26$, $p = 0.002$). Reference is made to Table 2.

### N1 amplitude and P2 amplitude

The grand average ERPs for the "Generate Tone" and "Hear Tone" conditions and the scalp topographies for the two conditions are presented in Figs 1 and 2, respectively. The mean N1 amplitude values for "Generate Tone" and "Hear Tone" are listed in Table 3. N1 amplitudes from Fz, FCz, and Cz for the two conditions in the two groups (patients with schizophrenia and control subjects) are illustrated in Fig 3. The results of the repeated measures ANOVA with condition ("Generate Tone" and "Hear Tone") and electrodes (Fz, FCz, Cz) as the between factors and group (patient, control) as the within factor are shown in Table 4.

According to the repeated measures ANOVA with Greenhouse-Geisser correction, the "Generate Tone" condition showed statistically significant N1 suppression relative to the "Hear Tone" condition, partial $\eta^2 = .47$. There was also a statistically significant interaction between the condition and group, partial $\eta^2 = .09$. The Bonferroni-adjusted post-hoc analysis of condition x group revealed a significantly higher N1 amplitude in patients with schizophrenia than in the control group for the "Generate Tone" condition ($M_{Diff} = 1.36$, $SD = .58$, 95% CI [.19, 2.52], $p = 0,023$); however, no significantly higher

**Table 1. Overview of the demographics.**

| Demographic | Schizophrenia Group M ± (SD) | Control Group M ± (SD) | p |
|---|---|---|---|
| Age (years) | 31.48 (7.69) | 30.67 (7.42) | 0.686 |
| Education (years) | 14.11 (3.58) | 16.22 (3.85) | **0.038** |
| Gender female/male (n) | 13/14 | 15/15 | 0.5 |
| PANSS | | | |
| Positive subscale (score) | 20.78 (7.57) | | |
| Negative subscale (score) | 17.52 (7.70) | | |
| General subscale (score) | 37.78 (10.66) | | |
| Total (score) | 75.19 (20.9) | | |
| Antipsychotic medication | | | |
| First generation only (n) | 0 | | |
| Second generation only (n) | 12 | | |
| Both (n) | 15 | | |
| Duration of the disease (years) | 7.74 (6.24) | | |

Overview of mean values (*M*) and standard deviations (*SD*) for age, gender, years of education, PANSS results, and type of medication. Significances (*p* < .05) are marked in bold.

**Table 2. Overview of subtests of the CCAS scale and WAIS-IV.**

| Subtests of the CCAS scale and WAIS-IV | Schizophrenia Group M ± (SD) | Control Group M ± (SD) | p |
|---|---|---|---|
| CCAS Semantic Fluency (Score) | 20.30 (7.42) | 26.73 (8.47) | **0.004** |
| CCAS Phonemic Fluency (Score) | 10.26 (3.95) | 14.4 (4.77) | **<0.001** |
| CCAS Category Switching (Score) | 11 (3.79) | 13.37 (3.82) | **0.023** |
| CCAS Digit Span Forward (Score) | 6.22 (1.12) | 6.33 (0.88) | 0.678 |
| CCAS Digit Span Backward (Score) | 4.22 (1.31) | 4.53 (0.86) | 0.29 |
| CCAS Cube (Draw) (Score) | 13.26 (3.19) | 14.4 (2.58) | 0.142 |
| CCAS Verbal Recall (Score) | 12.19 (2.88) | 13.7 (1.49) | **0.014** |
| CCAS Similarities (Score) | 7.11 (1.55) | 8 (0) | **0.003** |
| CCAS Go No-Go (Score) | 1.63 (0.69) | 2 (0) | **0.005** |
| CCAS Affect (Score) | 4.33 (1.31) | 5.97 (0.18) | **<0.001** |
| CCAS Total Failed Score | 2.63 (2.15) | 0.47 (0.86) | **<0.001** |
| CCAS Total Score | 91.26 (16.07) | 109.73 (15.25) | **<0.001** |
| WAIS-IV Raw Matrix Reasoning (Score) | 18.41 (4.87) | 21.43 (3.03) | **0.006** |
| WAIS-IV Raw Vocabulary (Score) | 40.89(12.73) | 49.47 (6.45) | **0.002** |

Results of the subtests of the CCAS scale and WAIS-IV. Mean values (*M*), standard deviations (*SD*), and significance values (*p*) of the various subtests of the CCAS scale and WAIS-IV were evaluated using unpaired *t*-tests. Significances (*p* < .05) are marked in bold.

N1 amplitude was observed in patients with schizophrenia compared with the control group for the "Hear Tone" condition ($M_{Diff}$ = −.496, $SD$ = .819, 95% CI [−2.14, 1.15], $p$ = .55). In summary, the control group showed a reduction in N1 amplitude during the "Generate Tone" condition, while patients with schizophrenia did not.

For the P2 amplitude, the mean amplitude values for the "Generate Tone" and "Hear Tone" conditions are listed in Table 5. P2 amplitudes from Fz, FCz, and Cz for the two conditions for patients with schizophrenia and control subjects are

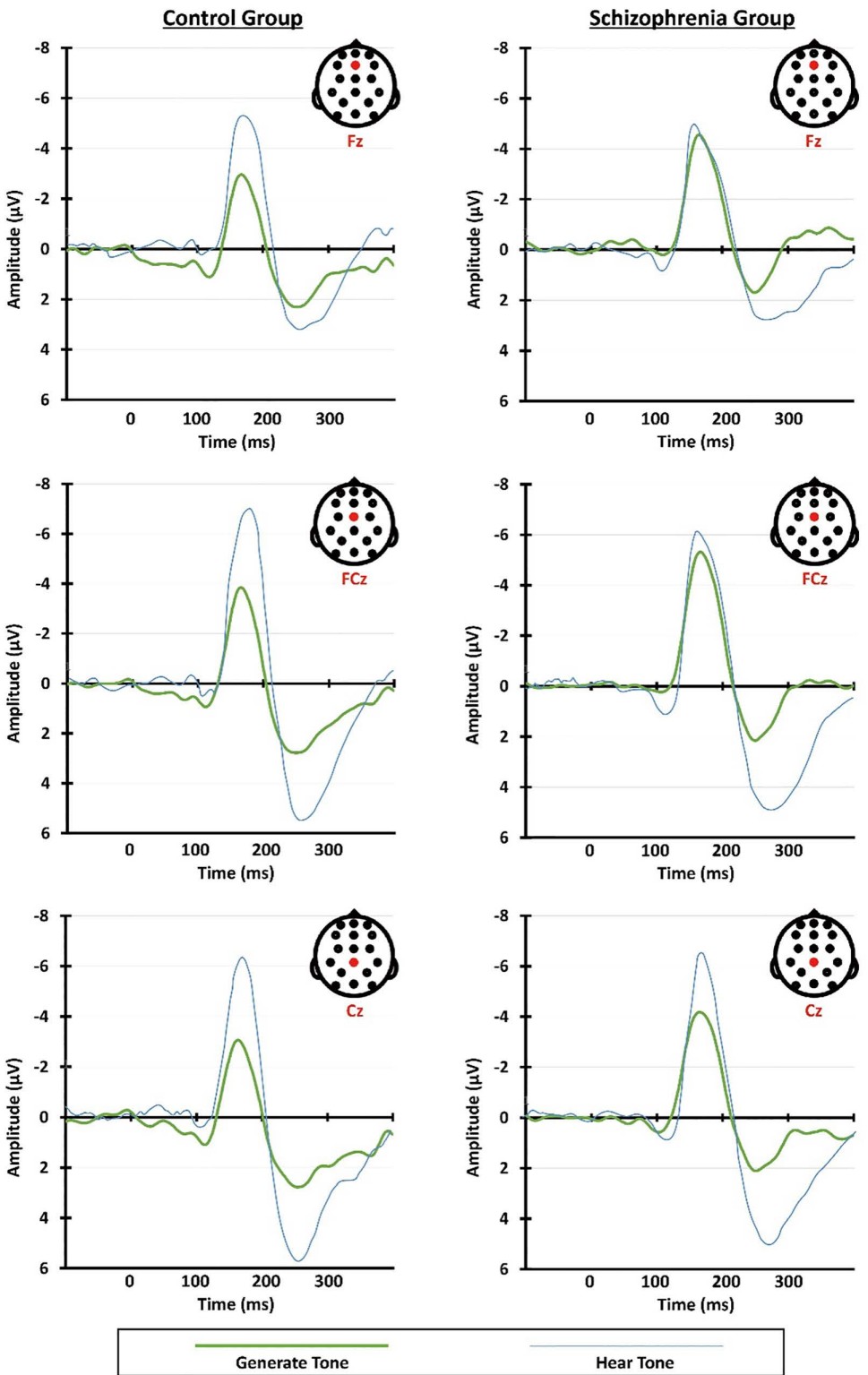

**Fig 1. Grand average event-related potentials of the N1 and P2.** Grand average event-related potentials of the N1 and P2 of patients with schizophrenia and the control group for the "Generate Tone" and "Hear Tone" conditions.

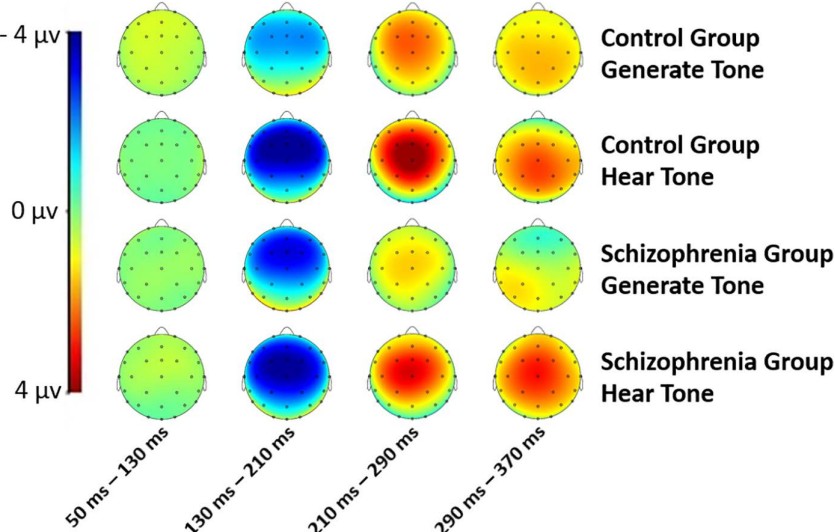

**Fig 2. Topographies of the scalps in the "Generate Tone" and "Hear Tone" conditions.** Topographies of the scalps of patients with schizophrenia and the control group in the "Generate Tone" and "Hear Tone" conditions.

**Table 3. Overview of the N1 amplitude values of patients with schizophrenia and the control group.**

|  | Fz N1 | | FCz N1 | | Cz N1 | |
| --- | --- | --- | --- | --- | --- | --- |
|  | Mean µV ± (*SD*) | | Mean µV ± (*SD*) | | Mean µV ± (*SD*) | |
|  | Generate Tone | Hear Tone | Generate Tone | Hear Tone | Generate Tone | Hear Tone |
| Schizophrenia Group | −5.15 (2.95) | −6.74 (3.57) | −5.83 (2.63) | −7.20 (3.97) | −4.76 (2.46) | −7.15 (3.81) |
| Control Group | −3.63 (2.17) | −6.97 (2.73) | −4.39 (2.13) | −8.45 (3.72) | −3.65 (2.16) | −7.38 (3.07) |

Mean values (*M*) and standard deviations (*SD*) of the N1 amplitude values of patients with schizophrenia and the control group in the "Generate Tone" and "Hear Tone" conditions.

displayed in Fig 4. The results of the repeated measures ANOVA with condition ("Generate Tone" and "Hear Tone") and electrodes (Fz, FCz, Cz) as the between factors and group (patient, control) as the within factor are shown in Table 6.

The repeated measures ANOVA with Greenhouse-Geisser correction of the P2 amplitude revealed a statistically significant difference between the "Generate Tone" and "Hear Tone" conditions, partial η²＝.425. There was no statistically significant interaction between group and condition, partial η²＝.001. Bonferroni-adjusted post-hoc analysis of the conditions revealed a significantly lower value of P2 in the "Generate Tone" condition than in the "Hear Tone" condition ($M_{Diff}$＝−2.967, *SD*＝0.465, 95% CI [−3.9, −2.03], *p*＜.001). In summary, the P2 amplitude decreased in the "Generate Tone" condition compared with the "Hear Tone" condition in both groups.

To evaluate the specificity of N1 and P2 results as peak to peak amplitude, we performed a repeated measures ANOVA with a Greenhouse-Geisser correction, including the within-factors ERP component ("N1 and P2"), condition ("Generate Tone" and "Hear Tone") and electrode (Fz, FCz, Cz), and group (patient, control) as between factors. The ANOVA results are presented in Table 7.

The repeated measures ANOVA with Greenhouse-Geisser correction of N1 and P2 yielded a statistically significant result for the interaction "N1 and P2" x condition, partial η²＝.62. We found no statistically significant interaction between "N1 and P2," condition and group, partial η²＝.03. Bonferroni-adjusted post-hoc analysis of the "N1 and P2" x condition

 

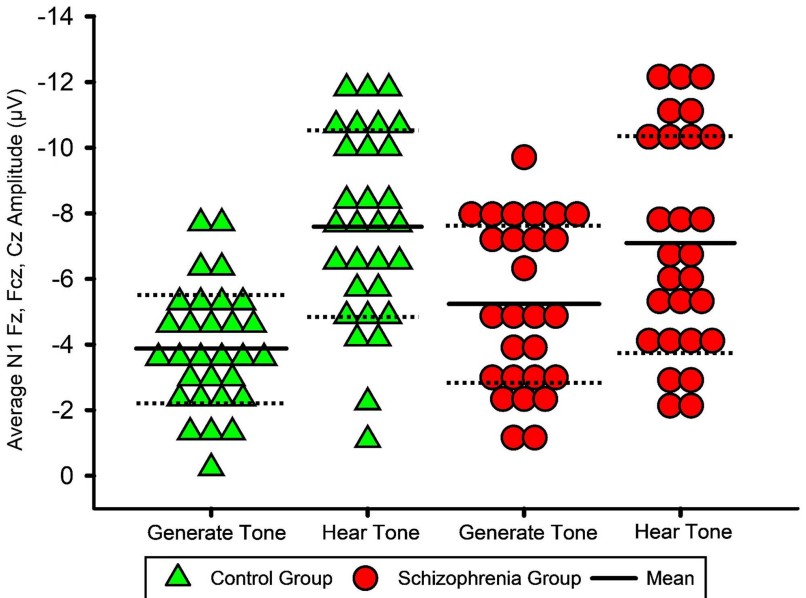

**Fig 3. Averaged N1 from Fz, FCz, and Cz for the "Generate Tone" and "Hear Tone" conditions.** Averaged N1 amplitudes from Fz, FCz, and Cz for the "Generate Tone" and "Hear Tone" conditions for the two groups: patients with schizophrenia and control subjects.

**Table 4. ANOVA with repeated measures for the N1 amplitude at electrodes Fz, FCz, and Cz.**

| ANOVA for N1 Amplitude at Electrodes Fz, FCz, and Cz | Df [1] | Df [2] | F | Significance Value |
|---|---|---|---|---|
| Group (CG vs SG) | 1 | 55 | 0.54 | 0.467 |
| **Condition (Generate Tone vs Hear Tone)** | **1** | **55** | **48.77** | **<0.001** |
| **Condition x Group** | **1** | **55** | **5.39** | **0.024** |
| **CG: Condition** | **1** | **29** | **57.79** | **<0.001** |
| **SG: Condition** | **1** | **26** | **8.37** | **0.008** |
| **Button Tone: Group** | **1** | **55** | **5.46** | **0.023** |
| Hear Tone: Group | 1 | 56 | 0.5 | 0.482 |
| **Electrodes (Fz, FCz, Cz)** | **1.50** | **82.22** | **6.78** | **0.004** |
| Electrodes x Group | 1.50 | 55 | 0.63 | 0.493 |
| Condition x Electrodes | 1.79 | 98.26 | 1.12 | 0.326 |
| Condition x Electrodes x Group | 1.79 | 55 | 1.49 | 0.232 |

ANOVA with repeated measures for the N1 amplitude at electrodes Fz, FCz, and Cz. Significances ($p < .05$) are in bold.

revealed a significant amplitude difference between N1 and P2 in the "Hear Tone" condition ($M_{Diff} = -13.740$, 95% CI [−15.11, − 12.37], $p < .001$). Similarly, we observed a significant difference in amplitude between N1 and P2 in the "Generate Tone" condition ($M_{Diff} = -7.99$, 95% CI [−8.82, −7.16], $p < .001$).

## Readiness potential

Fig 5 depicts the scalp topography of the patients with schizophrenia and subjects in the "Generate Tone" and "Button Alone" conditions. Fig 6 shows the Plot Groups – Averaged mean values and standard deviations of the electrodes Fz,

**Table 5. Overview of the P2 amplitude values of patients with schizophrenia and the control group.**

| | Fz P2 | | FCz P2 | | Cz P2 | |
| --- | --- | --- | --- | --- | --- | --- |
| | Mean µV ± (SD) | | Mean µV ± (SD) | | Mean µV ± (SD) | |
| | Generate Tone | Hear Tone | Generate Tone | Hear Tone | Generate Tone | Hear Tone |
| Schizophrenia Group | 2.54 (2.51) | 5.14 (4.95) | 2.91 (2.37) | 6.37 (3.85) | 3.08 (2.33) | 6.18 (3.02) |
| Control Group | 3.68 (1.80) | 5.52 (3.75) | 4.23 (1.86) | 7.57 (4.60) | 4.07 (1.73) | 7.54 (3.32) |

Mean values (M) and standard deviations (SD) of the P2 amplitude values of patients with schizophrenia and the control group in the "Generate Tone" and "Hear Tone" conditions.

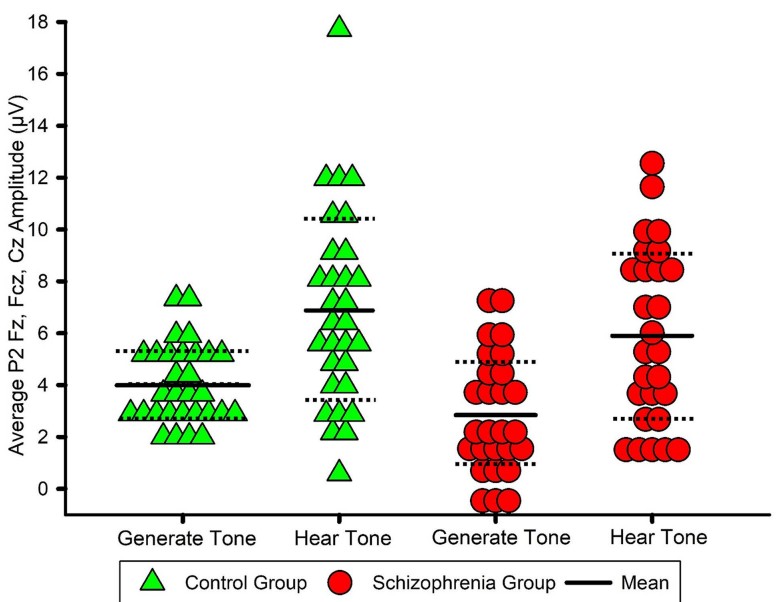

**Fig 4. P2 amplitudes at Fz, FCz, and Cz for the "Generate Tone" and "Hear Tone" conditions.** Averaged P2 amplitudes from Fz, FCz, and Cz for the "Generate Tone" and "Hear Tone" conditions for the two groups: patients with schizophrenia and control subjects.

**Table 6. ANOVA with repeated measures for the P2 amplitude at electrodes Fz, FCz, and Cz.**

| ANOVA for P2 Amplitude at Electrodes Fz, FCz, and Cz | Df 1 | Df 2 | F | Significance Value |
| --- | --- | --- | --- | --- |
| Group (CG vs SG) | 1 | 55 | 3.13 | 0.083 |
| **Condition (Generate Tone vs Button Alone)** | **1** | **55** | **40.64** | **<0.001** |
| Condition x Group | 1 | 55 | 0.04 | 0.852 |
| **Electrode (Fz, FCz, Cz)** | **1.67** | **91.56** | **11.74** | **<0.001** |
| Electrode x Group | 1.67 | 55 | 0.59 | 0.527 |
| **Condition x Electrode** | **1.55** | **85.27** | **4.73** | **0.018** |
| Condition x Electrode x Group | 1.55 | 55 | 0.91 | 0.385 |

ANOVA with repeated measurements for the P2 amplitude at electrodes Fz, FCz, and Cz. Significances (p < .05) are printed in bold.

**Table 7. ANOVA with repeated measures for N1 and P2 for electrodes Fz, FCz, and Cz.**

| ANOVA for N1 and P2 at Electrodes Fz, FCz, and Cz | Df ¹ | Df ² | F | Significance Value |
|---|---|---|---|---|
| **Group (CG vs SG)** | **1** | **55** | **4.51** | **0.038** |
| **N1 and P2** | **1** | **55** | **513.71** | **<0.001** |
| N1 and P2 x Group | 1 | 55 | 0.44 | 0.511 |
| Condition | 1 | 55 | 0.09 | 0.771 |
| Condition x Group | 1 | 55 | 2.64 | 0.110 |
| **Electrode** | **1.44** | **79.23** | **3.86** | **0.038** |
| **N1 and P2 x Condition** | **1** | **55** | **91.23** | **<0.001** |
| N1 and P2 x Condition x Group | 1 | 55 | 1.94 | 0.169 |

ANOVA with repeated measures for N1 and P2 for electrodes Fz, FCz, and Cz. Significances ($p < .05$) are printed in bold.

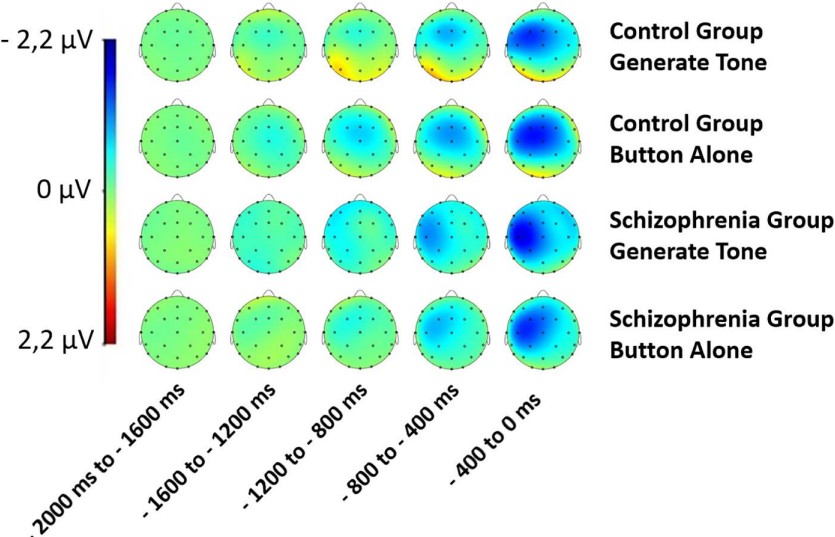

**Fig 5. Topographies of the scalps in the "Generate Tone" and "Button Alone" conditions before clicking.** Topographies of the scalps of patients with schizophrenia and the control group under the conditions "Generate Tone" and "Button Alone" before clicking.

FCz, Cz, C3, C4, F3 and F4 for all 3 conditions for patients with schizophrenia and control subjects in the time interval –2,000 ms to 0 ms in blocks of 200 ms each before clicking. Fig 7, in contrast, shows the plot by conditions. The ANOVA results are presented in Table 8.

A repeated measures ANOVA with Greenhouse-Geisser correction, with condition ("Generate Tone" and "Button Alone") and electrodes (Fz, FCz, Cz, C3, C4, F3, F4) as within-subject variables and group (patient with schizophrenia, control) as the between-subject factor, revealed a significant main effect of electrodes, partial $\eta^2 = .06$, divided segments, partial $\eta^2 = .37$, and an interaction effect of group x condition, partial $\eta^2 = .07$. No other significant effects or post-hoc effects were found in this analysis. A possible significant effect of N1 suppression was tested by including the conditional difference effect N1 ("Generate Tone") – N1 ("Hear Tone") in the ANOVA, but no relevant interactions were found.

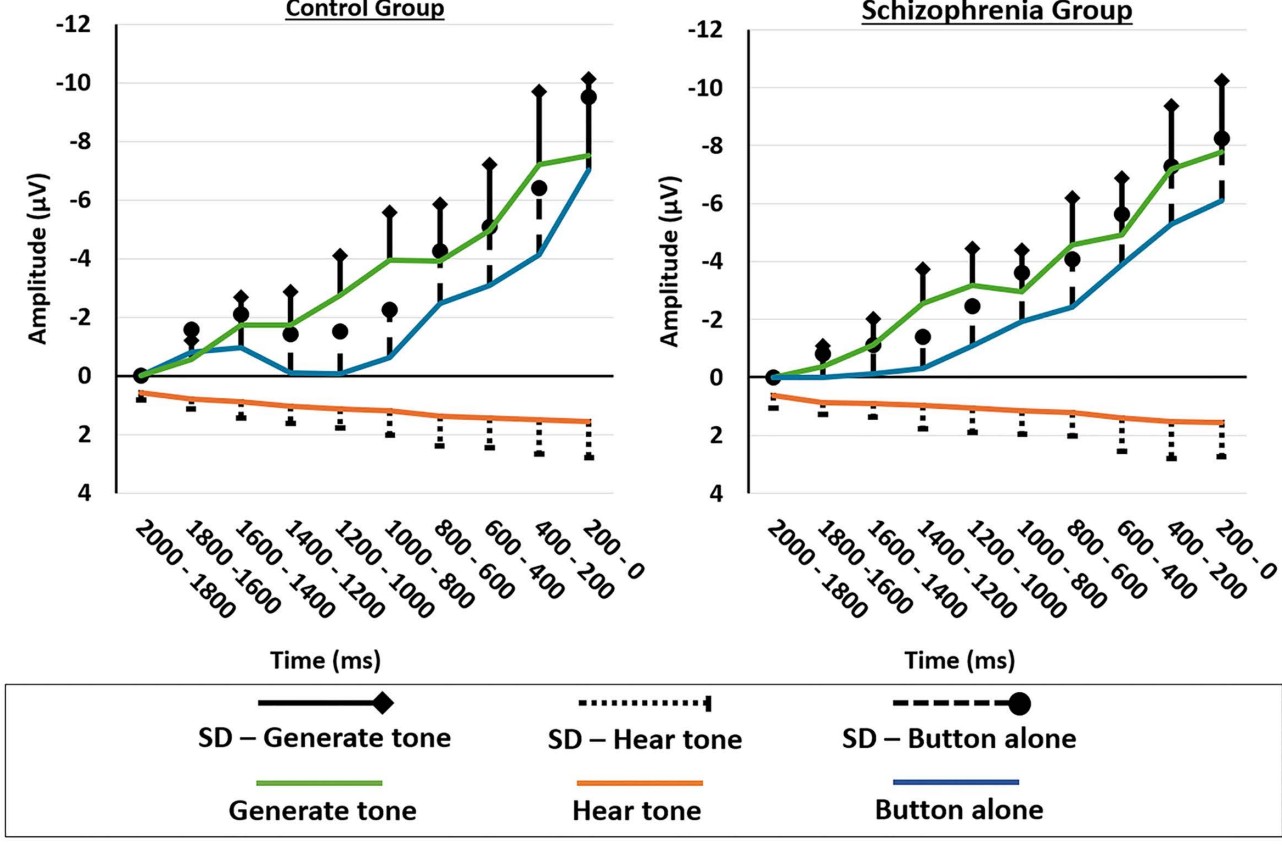

**Fig 6. Plot Groups – Averaged mean values and standard deviations of the readiness potential.** Averaged mean values and standard deviations of the readiness potential at the electrodes Fz, FCz, Cz, C3, C4, F3, and F4 for the conditions "Generate Tone", "Hear Tone", and "Button Alone" for patients with schizophrenia and subjects in the time interval 2,000 ms to 0 ms of 200 ms each before clicking.

### ERP amplitudes vs RP and LRP amplitude

To test the relationship between N1 and P2 on the one hand and RP and LRP on the other, we carried out two measurements in preparation for the analysis. First, we calculated the difference between N1 and P2 in the electrode FCz under the "Generate Tone" and "Hear Tone" conditions. Second, we calculated the RP difference in the time interval from −400 ms to −200 ms before the click of a tone in electrodes C3 and C4 under the "Generate Tone" condition. The differences were then calculated using Pearson's correlation coefficient. The regressions were carried out both separately for group membership and without considering group membership as a factor. No significant results or correlations were found in either the group-dependent or group-independent analyses.

Moreover, we conducted a MANOVA for the RPs and C3-C4 LRPs including the three conditions and 10 segments. We found no effect of group factor for RP ($F(1, 55) = 0,03$, $p = 0.863$) or LRP ($F(1, 55) = 3.35$, $p = 0.073$). Also, there was not a statistically significant interaction between the condition and group for RP ($F(1.92, 105.83) = 2.19$, $p = 0.119$) or LRP ($F(1.96, 107.74) = 0.57$, $p = 0.563$). In addition to C3-C4 LRPs we assessed probable associations of condition "Generate Tone" and "Hear Tone" with covariance to the difference of the N100 component. A further covariance analysis included the C3-C4 segment from −400–200 ms prior to button press with the "Generate Tone" and "Hear Tone" conditions to analyze possible associations with the lateralized readiness potential or event-related potential. These two analyses yielded no effects involving the group factor.

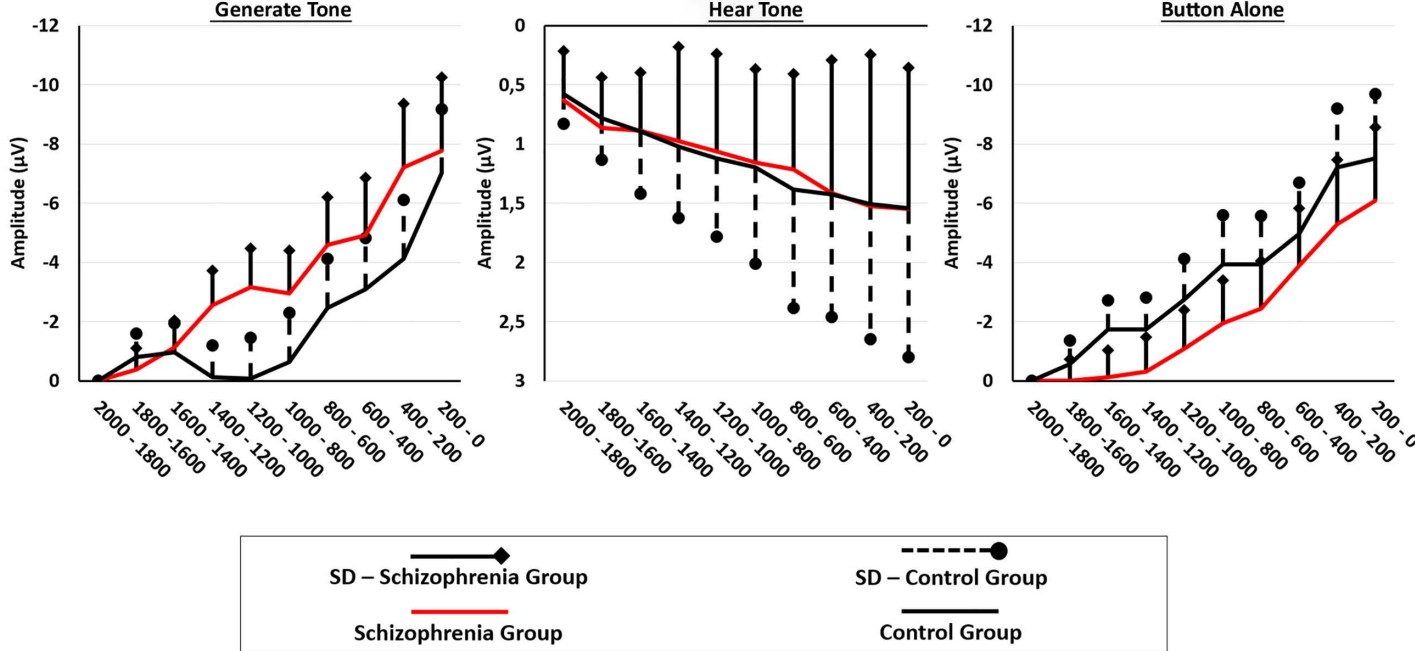

**Fig 7. Plot Conditions – Averaged mean values and standard deviations of the readiness potential.** Averaged mean values and standard deviations of the readiness potential at the electrodes Fz, FCz, Cz, C3, C4, F3, and F4 for patients with schizophrenia and subjects for the conditions "Generate Tone", "Hear Tone", and "Button Alone" in the time interval 2,000 ms to 0 ms of 200 ms each before clicking.

**Table 8. ANOVA with repeated measures for the readiness potential.**

| ANOVA for Readiness Potential | Df [1] | Df [2] | F | Significance Level |
|---|---|---|---|---|
| Group (CG vs SG) | 1 | 55 | 0.02 | 0.896 |
| Condition (Generate Tone vs Button Alone) | 1 | 55 | 0.01 | 0.908 |
| **Condition x Group** | **1** | **55** | **4.40** | **0.041** |
| SG: Condition | 1 | 26 | 2.68 | 0.113 |
| CG: Condition | 1 | 29 | 2.03 | 0.165 |
| Generate Tone: Group | 1 | 55 | 2.23 | 0.141 |
| Button Alone: Group | 1 | 55 | 1.41 | 0.241 |
| **Divided Segments** | **2.72** | **149.52** | **33.30** | **<0.001** |
| Divided Segments x Group | 2.72 | 55 | 0.34 | 0.78 |
| Condition x Divided Segments x Group | 2.83 | 55 | 1.83 | 0.143 |
| **Electrodes** | **4.80** | **263.86** | **3.59** | **0.004** |
| Electrode x Group | 4.80 | 55 | 1.36 | 0.242 |
| Condition x Electrode x Group | 4.37 | 55 | 1.05 | 0.385 |

ANOVA with repeated measures for the readiness potential for electrodes Fz, FCz, Cz, C3, C4, F3, and F4. Significances ($p < .05$) are marked in bold.

## Button pressing pace and average number of included trials

We found no significant relationships between button pressing pace and average number of included trials.

## Relationship between clinical symptoms and ERP measures

We found no statistical correlation between clinical symptoms and N1 and P2 ERP measures.

### Relationship between PANSS and cognitive performance

We found a significant negative correlation between negative symptoms according to the PANSS and the subtests "Semantic word fluency" ($r = -.482$, $p = .011$), "Phonemic word fluency" ($r = -.648$, $p < .001$), and "Category change" ($r = -.420$, $p = .029$) from the CCAS scale. In addition, we observed a significant correlation between negative symptoms and "total number of points achieved" ($r = -.607$, $p < .001$) and "total number of all failed individual tests" ($r = .446$, $p = .02$).

### Relationship between illness duration and cognitive performance

We found a significant negative correlation between illness duration and the subtest "Semnatic word" ($r = -.386$, $p = .05$) from the CCAS scale. There was no significant correlation between duration of illness and "total number of points achieved" ($r = -.329$, p = .09) or "total number of all failed individual tests" ($r = .341$, p = .08).

## Discussion

The first main finding of our study is the absence of N1 amplitude suppression in patients with schizophrenia when presented with self-initiated auditory stimuli, compared with healthy subjects. This interaction effect was demonstrated when evaluating the three electrodes (Fz, FCz, and Cz) for the "Generate Tone" and "Hear Tone" conditions in both the SG and CG groups. Further analyses revealed no significant difference between groups in the "Hear Tone" condition, indicating that auditory perception remained unchanged. However, a statistically significant group difference was found in the "Generate Tone" condition, suggesting that patients' CD is impaired at the onset of sensory input. To the best of our knowledge, there has been no exact replication of the experimental setup of Ford et al.'s study [32] with self-initiated tone stimuli and button presses as self-initiated actions. However, studies exist that demonstrate the deficits of the CD mechanisms in schizophrenia patients using different stimuli and actions [9,13–17]. The exact underlying mechanisms are still under investigation and remain controversial [44]. Nevertheless, numerous studies have suggested that the interplay between EC and CD is critical for effective sensorimotor integration [9,32,45,46]. If this model is correct, then imposing a delay between the auditory-evoking action and the subsequent auditory feedback might explain subnormal levels of N1 suppression in patients with schizophrenia. This in turn suggests that forward modeling abilities may be impaired [7], and failure to predict self-generated sensations has been suggested to be associated with the experience of auditory verbal hallucinations. However, this theory requires further investigation, as it is limited by a lack of studies. In conclusion, our results replicate the primary findings of Ford et al. [32] and show that the N1 to auditory stimuli is not impaired in healthy controls but that the information processing deficits in schizophrenia become evident in the case of sensory prediction processes. Furthermore, our results are consistent with existing evidence of sensory prediction deficits in schizophrenia patients, reported using different experimental methods and sensory modalities.

Furthermore, our study also found a considerable decrease in P2 activity in both groups during the "Generate Tone" condition rather than the "Hear Tone" condition. Notably, Ford's study [32] did not show significant suppression of P2 in either group for the "Generate Tone" condition compared with the "Hear Tone" condition. In agreement with Ford's results, we did not observe any interaction between condition and group. Furthermore, our results suggest that there may be an interaction between the N1 and P2 components in response to the two conditions. However, the interaction term in the concurrent analysis of N1 and P2, which included the group factor, did not prove to be statistically significant. In order to achieve statistical power for this effect, a larger sample size will be needed. Nevertheless, our data and the significant component x condition interaction suggest that both components react differently to the two conditions, possibly due to varying CD mechanisms. To our opinion, this indicates that the P2 component should be considered in more depth in the assessments of basic CD mechanisms [44]. If the results are compared, it is crucial to note, as discussed [47], that N1 and P2 suppression effects show different responses to different experimental manipulations. For example, one study

revealed P2 differences by using different methods to perform the action (hand versus foot) [14], and another study found that the length of time between the action and the produced sound can also produce different effects on N1 and P2 suppression [47]. Given the varying N1 and P2 suppression effects in response to various experimental manipulations, it seems possible that these two components reflect distinct mechanisms that are functionally dissociable [47]. Alternatively, these two components may represent different steps along a common chain of processes related to self-generation [47,48]. In short, the number of studies addressing P2 in general and in the context of schizophrenia remains limited [49], thus leaving uncertainty about the most likely representation of P2.

The second aim, analysis of the RP data in the two conditions ("Generate Tone" and "Button Press") and groups ("Patients" and "Controls") revealed a marginally significant interaction effect. In controls, the RP in the "Generate Tone" condition showed a lately negative amplitude shift compared with the RP in the "Button Alone" condition. In patients, the "Button Alone" condition showed a small but consistent negative potential shift in the preceding the motor action compared to the "Generate Tone" condition. Subsequent analysis of the condition effect in both groups separately and of the group effects in the experimental conditions revealed no significant effects, suggesting that the significant interaction effect is evidently only in the combined analysis of groups and conditions.

Previous studies have shown that the primary sensorimotor cortex, supplementary motor area (SMA), premotor cortex, and prefrontal cortex are involved in the generation of RP [50–53]. Hence, changes in early RP may result from changes in the SMA, which is involved in movement preparation [54], and effective connectivity between these areas appears to be critical for the maintenance of RP [55]. Several studies have observed a correlation between motor dysfunction and schizophrenia in the sensorimotor cortex and SMA [56,57]. However, Horváth [44] highlights an important consideration: The act of moving a finger to press a button involves neural processes that are responsible for the preparation and execution of the action. It has been debated whether the sensory suppression observed is caused by movement planning, movement execution, or both. Jack et al. attempted to address this question [58], examining the effects of both involuntary and semi-voluntary movements on N1 responses. The experimenter induced the involuntary movements by moving the participant's finger or stimulating the median nerve, while the semi-voluntary motions involved the participant using a finger from their left hand to activate a button on their right hand or using their left hand to activate a nerve stimulator placed on their right arm. Interestingly, the authors found no significant difference in the N1 responses to tones resulting from semi-voluntary or involuntary movements when compared with passively heard tones. These findings align with the forward-model theories of action prediction and indicate that motion planning, which creates an EC of the predicted sensory action consequences, is imperative for generating neural sensory inhibition in the auditory modality. It would be worthwhile to replicate this study with patients who have schizophrenia to further expand on the hypothesis of the forward model.

Regarding the relationship between N1 and LRP, Ford [32] demonstrated that participants with a larger LRP showed greater suppression of N1, suggesting that the correlation between N1 and LRP reflects the generation of EC. However, our results are inconsistent with Ford's findings, considering that we analyzed 200-ms mean amplitude blocks starting at 2,000 ms before button press and lasting from onset to button press. This method was also used in previous studies conducted by our team [59,60]. In our experiment, the completion of the mouse click required minimal motor effort, whereas increased motor demands would yield larger amplitudes of RP [61]. In parallel to this, there have been other recent research and papers on corollary discharge mechanisms that also show the opposite [62–66]. We addressed this issue and performed extended control analyses using RP/LRP ANOVA with all 3 conditions and the addition of covariances. Again, no effects were found. Overall, it remains to be seen whether RP/LRP can function as an indicator of CD signaling.

In addition to the question of the exact underlying mechanism, a further question concerns the origin of CD impairment. Whitford hypothesized that damage to white matter (WM) fibers, which serve as a transport pathway for volitionally initiated CD, may be the source of deficits in "self-monitoring" [67]. WM consists primarily of myelinated axon bundles that

increase the speed of action potential conduction through electrically insulating axons. Damage to the WM typically results in conduction delays [68,69]. Therefore, Whitford's group – based on studies investigating abnormalities in patients with schizophrenia, either post-mortem using microscopy [69] or in vivo with structural MRI [70] – suggested that as a result of demyelination a delay occurs in the EC, which in turn leads to symptoms of schizophrenia. In summary, the observation of an RP difference in healthy individuals versus in those with schizophrenia was evident in this study and is consistent with findings in previous studies [26–31]. This distinction was particularly pronounced during the initial stages of tone generation in healthy individuals. Additionally, no correlation was observed between the suppression of N1 and LRP, which may again be attributed to the experimental design and analysis methods employed. Further research is necessary to examine this association.

As a final aim of this study, we replicated previous findings on impaired cognitive performance in patients with schizophrenia, focusing on executive function [37,71]. Furthermore, a significant negative correlation could be demonstrated between negative symptoms and part of the CCAS scale tests, as has also been shown in previous work [72]. Moreover, we found no association between ERP measures and cognition tests. It should be noted, however, that Mathalon and Ford [73] have already indicated, despite advances in neuroimgaging and other methods, the challenges of linking sensory suppression to the clinical symptoms of schizophrenia remain. Mathalon and Ford suggest several obstacles that might contribute to this lack of correlation, including small sample sizes, unreliable measures and medication effects. These factors prevent a clear understanding of how specific neurobiological abnormalities relate to the various clinical symptoms of schizophrenia. In the discussion of the relationship between illness duration and cognitive performance, a negative correlation was also found between illness duration and the subtest 'Semantic Word Fluency'. In addition, a negative correlation between illness duration and cognition was also observed that just failed significance. It is important to note that the present study exclusively included patients within the age range of 18–45 years, and the analysis was further refined by considering the small patient sample for such studies. Nevertheless, other studies have shown that illness Duration is associated with progressive decline in both cognitive and cerebellar function, highlighting the role of disease progression in causing further neurocognitive impairment [74,75].

To assess cognitive performance, we used two subtests of the Wechsler Adult Intelligence Scale (WAIS-IV), namely matrix reasoning and verbal comprehension, which focus on logical reasoning and verbal skills, respectively. By contrast, the CCAS scale is a new compilation of classic executive function tests by Schmahmann and colleagues [39]. Schmahmann described lesions in the cerebellum of the brain as cerebellar cognitive-affective syndrome, which refers to a constellation of deficits in high-level functions [76]. While the subtests of the CCAS have long been known to be sensitive to executive function [77], it has recently become clear that standard executive functions depend on intact cerebellar function [78–80]. The study by Knolle [81] that highlights the absence of N1 suppression in individuals with cerebellar lesions is notable. In addition to prior research demonstrating the participation of the cerebellum in the EC/CD mechanism and its integration into the forward model [82], this study also examines evidence from two decades suggesting that schizophrenia is associated with cerebellar abnormalities. Historically, the cerebellum has been thought to be primarily responsible for motor coordination [83]. In recent years, however, several studies have suggested that the cerebellum is not only involved in the regulation of movement but also plays a role in advanced cognitive and emotional processes [84]. Dysfunctional connectivity between the cortex, cerebellum, pons, and thalamus has been recognized [85–89], and reduced cerebellar gray matter volume has been reported in schizophrenia [90]. In addition, an attempt has been made to treat schizophrenia by stimulating the cerebellum [91]. Another theoretical account points to myelin abnormalities in the cerebellum as described above, which may disrupt CD signaling, according to a theoretical hypothesis by Whitford [67]. To our knowledge, this is the first time CCAS-Scale has been used in patients with schizophrenia. Therefore, we added the CCAS to the study to emphasize the importance of the cerebellum, which has received increased research attention in recent years. However, empirical evidence is still lacking. In conclusion, the cerebellum may contribute to the function and dysfunction of CD mechanisms in schizophrenia.

## Limitations and strengths

Although the sample size of the study was limited and recruitment was difficult, particularly due to the COVID-19 pandemic, it was still adequate for detecting significant results. In addition, the patients with schizophrenia in our study had higher PANSS scores than those in the Ford study, especially on the positive scale and the general scale, which could bias the comparison of the studies [32].

In addition, when considering the correlation between N1 and LRP, as mentioned in the Discussion section [61], it must be taken into account that a larger readiness potential amplitude may occur depending on the force required to click the mouse, which in turn could make it difficult to compare study results. For this reason, we would recommend that follow-up research include the force required to press a button, which could increase the comparability between studies.

## Conclusion

In conclusion, by replicating Ford et al.'s (2014) observations, we provide evidence that pressing a key to deliver a tone produces a similar pattern of findings: When an auditory stimulus is evoked by button presses, patients with schizophrenia exhibit a lack of suppression of N1, but not the P2 component. This indicates a self-suppression deficit in schizophrenia regarding the N1 component. Another observation is that RP differs in the interaction of conditions and groups. Taken together, these findings support the idea that schizophrenia is most likely related to physiological abnormalities in CD/EC [5,6,92]. Furthermore, our neurocognitive assessment covering probable sensitivity to cerebellar function has shown relevance in the context of schizophrenia.

## Supporting information

**S1 File.** **https://duepublico2.uni-due.de/receive/duepublico_mods_00084480**
(DOCX)

## Acknowledgments

The authors would like to thank M.Sc., Psych., Anna-Maria Reinartz for her competent help with data acquisition and pre-processing. This manuscript was written as part of the medical degree thesis by author CLK.

## Author contributions

**Conceptualization:** Emma Sprooten, Bernhard W. Müller.

**Data curation:** Constantin Liermann-Koch.

**Formal analysis:** Constantin Liermann-Koch.

**Funding acquisition:** Norbert Scherbaum, Emma Sprooten.

**Investigation:** Constantin Liermann-Koch, Jan W. Thielen, Dae-In Chang, Richard Krieger-Strásky, Oliver Kraff.

**Methodology:** Bernhard W. Müller.

**Project administration:** Bernhard W. Müller.

**Resources:** Jan W. Thielen, Dae-In Chang, Richard Krieger-Strásky, Oliver Kraff, Norbert Scherbaum, Emma Sprooten.

**Software:** Bernhard W. Müller.

**Supervision:** Bernhard W. Müller.

**Validation:** Norbert Scherbaum.

**Visualization:** Constantin Liermann-Koch.

**Writing – original draft:** Constantin Liermann-Koch.

**Writing – review & editing:** Jan W. Thielen, Dae-In Chang, Richard Krieger-Strásky, Oliver Kraff, Norbert Scherbaum, Emma Sprooten, Bernhard W. Müller.

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
