## [Decision Letter · Decision Letter 0]

18 Feb 2025

PONE-D-24-58770Corollary discharge and efference copy mechanisms in schizophrenia and controls: The N1 and P2 evoked potential components differentially react to self-initiated tones in schizophreniaPLOS ONE

Dear Dr. Liermann-Koch,

Thank you for submitting your manuscript to PLOS ONE. After careful consideration, we feel that it has merit but does not fully meet PLOS ONE’s publication criteria as it currently stands. Therefore, we invite you to submit a revised version of the manuscript that addresses the points raised during the review process.

We look forward to receiving your revised manuscript.

Kind regards,

Jie Wang, Ph.D.

Academic Editor

PLOS ONE

2. Thank you for stating the following financial disclosure:  [This study was partly funded by a NARSAD Young Investigator Award of the Brain and Behavior Research Foundation (BBRF), 90 Park Avenue, 16 th floor, New York, NY, USA. for P.I. Emma Sprooten, Ph.D; Grant ID: 25034].  Please state what role the funders took in the study.  If the funders had no role, please state: "The funders had no role in study design, data collection and analysis, decision to publish, or preparation of the manuscript." If this statement is not correct you must amend it as needed. Please include this amended Role of Funder statement in your cover letter; we will change the online submission form on your behalf.

3. In the online submission form, you indicated that [Original data may be provided upon resonable request.]. All PLOS journals now require all data underlying the findings described in their manuscript to be freely available to other researchers, either 1. In a public repository, 2. Within the manuscript itself, or 3. Uploaded as supplementary information.This policy applies to all data except where public deposition would breach compliance with the protocol approved by your research ethics board. If your data cannot be made publicly available for ethical or legal reasons (e.g., public availability would compromise patient privacy), please explain your reasons on resubmission and your exemption request will be escalated for approval.

Additional Editor Comments:

The authors need to address the comments raised by the two reviewers.

Reviewers' comments:

Reviewer's Responses to Questions

**Comments to the Author**

1. Is the manuscript technically sound, and do the data support the conclusions?

Reviewer #1: Yes

Reviewer #2: Yes

2. Has the statistical analysis been performed appropriately and rigorously? 

Reviewer #1: Yes

Reviewer #2: Yes

3. Have the authors made all data underlying the findings in their manuscript fully available?

Reviewer #1: No

Reviewer #2: No

4. Is the manuscript presented in an intelligible fashion and written in standard English?

Reviewer #1: Yes

Reviewer #2: Yes

5. Review Comments to the Author

Reviewer #1: Liermann-Koch and colleagues have presented a replication of an important study (Ford et al. 2014) into sensorimotor prediction in patients with Schizophrenia and healthy controls. They have collected a very reasonable sample size and the paper is well-written. I would like to suggest some further analyses which will help to bring the paper in line with current research that has taken place since Ford et al (2014) was published. I hope the authors will agree that these suggestions will increase the paper’s reach and will be particularly valuable to researchers in the motor control field, especially given the rarity of this type of patient data. I will also make some minor suggestions.

Major revisions:

Although Ford et al. (2014) showed a correlation between LRP and N1, several other papers have conversely implicated readiness potential in action-effect prediction (Ody et al., 2023; Pinheiro, Schwartze, Amorim, et al., 2020; Pinheiro, Schwartze, Gutiérrez-Domínguez, et al., 2020; Reznik et al., 2018; Vercillo et al., 2018). RP and LRP are thought to represent different processes, with RP being sensitive to higher-level motor preparation (for timing, trajectory, predicting sensory consequences etc.) and LRP being related to lower-level motor-specific processes (i.e. preparing to execute the action itself). As these data are readily available, the authors have the opportunity to provide some valuable further analysis to answer the following questions: A) Does the RP/LRP specifically encode the action’s outcome or general anticipation of an upcoming stimulus (see (Reznik et al., 2018; Vercillo et al., 2018)) and B) does this process differ between patients and controls? I suggest including all three conditions in the analysis for RP. This would allow one to answer whether RP is specific to motor prediction, or reflects a more general anticipation process. Secondly, I suggest repeating the ANOVA for the LRP. If RP and LRP reflect different processes, we would expect to see that in these two analyses. Finally, it would be great to see the RP and LRP waveforms plotted for all conditions and for both groups, to get a clearer picture of the overall results. I believe these wouldn’t go outside the paper’s intention and would be valuable results to have in a rare sample.

Minor revisions:

The results section is a bit confusing. Generally, if the results are reported in the text, they don't need to be reported in a table. I think it would be more effective to report everything in the tables, including both types of degrees of freedom and the effect sizes and then highlight the significant results in the text, for example, ‘As shown in Table 2, there was a significant difference between X and Y, with higher amplitudes in X (estimated marginal mean = xyz) than Y (EMM = xyz).’ Otherwise, please consider consolidating the results in another way to make them easier to read.

After reading the RP sections in the methods and results, it wasn't clear which time window was analysed. In the methods (lines 255-256), it suggests multiple analyses were conducted. However, in the results, only one value per analysis is given, so I suspect there must be a single selected time window. If this is the case, could the authors please clarify how this window was chosen?

For the ERP analyses, what is the reason for including electrodes as a factor, rather than taking an average across them? Do the authors have a reason to believe that N1/P2/RP would differ between electrodes? If not, I recommend running the analysis on an average of the selected electrodes, as this would make the results easier to parse.

There is a possible error in the text in lines 408/409, where it lists the included conditions as 'generate tone and 'hear tone'. I understand from the earlier text that this analysis included the 'generate tone' and 'button alone' conditions. However, see my above comment regarding including all three conditions.

For Fig 3 and 4, it would be better to see the individual data points (see, for example, (Rousselet et al., 2016)). This technique is more effective than a simple error bar at showing the dispersion of the data.

The authors might consider including a brief discussion regarding the lack of correlation between sensory suppression and clinical symptoms. This has been a consistent difficulty for the field that some consider to be a key piece of lacking evidence for connecting deficits in the forward model to reduced sensory suppression in patients. (Mathalon & Ford, 2012) gives a detailed discussion of this issue.

A couple of minor corrections to the text:

Line 127 says the study (Roach, 2023) was ‘aforementioned’, but I think it wasn’t mentioned before this sentence.

In line 246, there is an extra quotation mark before “Button Alone”.

Mathalon, D. H., & Ford, J. M. (2012). Neurobiology of schizophrenia: search for the elusive correlation with symptoms. Frontiers in Human Neuroscience, 6, 136.

Ody, E., Kircher, T., Straube, B., & He, Y. (2023). Pre-movement event-related potentials and multivariate pattern of EEG encode action outcome prediction. Human Brain Mapping, 44(17), 6198–6213.

Pinheiro, A. P., Schwartze, M., Amorim, M., Coentre, R., Levy, P., & Kotz, S. A. (2020). Changes in motor preparation affect the sensory consequences of voice production in voice hearers. Neuropsychologia, 146(January), 107531.

Pinheiro, A. P., Schwartze, M., Gutiérrez-Domínguez, F., & Kotz, S. A. (2020). Real and imagined sensory feedback have comparable effects on action anticipation. Cortex; a Journal Devoted to the Study of the Nervous System and Behavior, 130, 290–301.

Reznik, D., Simon, S., & Mukamel, R. (2018). Predicted sensory consequences of voluntary actions modulate amplitude of preceding readiness potentials. Neuropsychologia, 119(January), 302–307.

Rousselet, G. A., Foxe, J. J., & Bolam, J. P. (2016). A few simple steps to improve the description of group results in neuroscience. European Journal of Neuroscience, 44(9), 2647–2651.

Vercillo, T., O’Neil, S., & Jiang, F. (2018). Action-effect contingency modulates the readiness potential. NeuroImage, 183, 273–279.

Reviewer #2: I appreciate the opportunity to review this valuable paper.

This paper is a comprehensive replication and validation of Ford's (2014) study. It is a thought-provoking paper on the origins and research of corollary discharge and is worthy of inclusion. However, there are several points that need to be clarified by the authors.

Major points

#1 The patient population in this study was diverse, with 14 diagnoses of paranoid schizophrenia, seven of undifferentiated schizophrenia, three of residual schizophrenia, three of schizoaffective disorder and one of hebephrenic schizophrenia. The mean duration of illness is given as 7.74 years, but the range is unclear and should be clarified.

In other words, the study includes a diverse group of patients with different backgrounds. It is assumed that cerebellar and cognitive decline progresses with disease duration. The relationship between disease duration and cognitive function should be discussed.

#2 As the author states in the first paragraph of the Discussion, it has been suggested that changes in CD are associated with positive symptoms in particular. However, the paper does not address the association between positive symptoms on the PANSS and ERPs, which may be representative of changes in CD. If the results did not demonstrate an association, this would still need to be addressed in the discussion. Related to the above point #1, it may be that schizophrenia is thought to be preceded by positive or negative symptoms, and that cognitive decline becomes more pronounced as the illness progresses, but these points also need to be considered.

Minor points

#3

Title of Table 5 and 6; not “N1” but “P2”.

#4

P13, L369; Does ERP component ("N1 and P2") mean peak-to-peak amplitude? I did not understand this procedure, although I am not familiar with this description.

#5

P17, L501; "Changes in early RP may result from structural changes in the SMA..."

My understanding is that the term "structural" means more visible changes on MRI or other imaging. Perhaps the authors consider these changes to be permanent rather than transient, and therefore use the term "structural". However, I wonder if this word might be misleading.

6. PLOS authors have the option to publish the peer review history of their article (what does this mean? ). If published, this will include your full peer review and any attached files.

**Do you want your identity to be public for this peer review?** For information about this choice, including consent withdrawal, please see our Privacy Policy .

Reviewer #1: **Yes: ** Edward Ody

Reviewer #2: **Yes: ** Kazuyori Yagyu

---

## [Author Response · Author response to Decision Letter 1]

10 Jul 2025

Dear colleagues from PLOSone, Dr. Ody and dear Prof. Kazuyori Yagyu,

as part of the review process, I would like to thank you very much for the prompt processing of the manuscript and the very good suggestions made by the two reviewers, so that I was able to expand the manuscript very positively from my point of view during the re-editing. For better readability, I have created a Word file with the name “Questions and Answers” in which I answer the individual comments of the two reviewers. Please note that when quoting the page and line I refer to the document “Manuscript with marked changes”.

In the following sections, I will provide feedback on each comment made by the two reviewers.

Yours sincerely,

Constantin Liermann-Koch

#Reviewer 1:

Major issues

1# Comment: Although Ford et al. (2014) showed a correlation between LRP and N1, several other papers have conversely implicated readiness potential in action-effect prediction (Ody et al., 2023; Pinheiro, Schwartze, Amorim, et al., 2020; Pinheiro, Schwartze, Gutiérrez-Domínguez, et al., 2020; Reznik et al., 2018; Vercillo et al., 2018). RP and LRP are thought to represent different processes, with RP being sensitive to higher-level motor preparation (for timing, trajectory, predicting sensory consequences etc.) and LRP being related to lower-level motor-specific processes (i.e. preparing to execute the action itself). As these data are readily available, the authors have the opportunity to provide some valuable further analysis to answer the following questions: A) Does the RP/LRP specifically encode the action’s outcome or general anticipation of an upcoming stimulus (see (Reznik et al., 2018; Vercillo et al., 2018)) and B) does this process differ between patients and controls? I suggest including all three conditions in the analysis for RP. This would allow one to answer whether RP is specific to motor prediction, or reflects a more general anticipation process. Secondly, I suggest repeating the ANOVA for the LRP. If RP and LRP reflect different processes, we would expect to see that in these two analyses. Finally, it would be great to see the RP and LRP waveforms plotted for all conditions and for both groups, to get a clearer picture of the overall results. I believe these wouldn’t go outside the paper’s intention and would be valuable results to have in a rare sample.

1# Answer: The extended analysis of the 3 conditions at RP and LRP including covariances Difference of the N100 component and C3-C4 Segment from -400 to 200 ms prior to button press with the “Generate Tone” and “Hear Tone condtions” was performed. Overall, there was no significant result with regard to the group factor. This static analysis was described including the results on page 15, lines 433 – 440. The section is attached here:

Moreover, we conducted a MANOVA for the RPs and C3-C4 LRPs including the three conditions and 10 segments. We found no effect of group factor for RP (F(1.92, 105.83) = 2.19, p = 0.0116) or LRP (F(1.96, 107.74) = 0.57, p = 0.563). In addition to C3-C4 LRPs we assessed probable associations of condition “Generate Tone” and “Hear Tone” with covariance to the difference of the N100 component. A further covariance analysis included the C3-C4 segment from -400 to 200 ms prior to button press with the “Generate Tone” and “Hear Tone” conditions to analyze possible associations with the lateralized readiness potential or event-related potential. These two analyses yielded no effects involving the group factor.

We also processed the readiness potential graphs by status and added the figures sorted by condition. During this process, I noticed that the original graph for readiness potential was not prepared correctly and the graph was split by condition instead of status. The graphic fig. 6 and description (page 17, line 515 - 519) have been adjusted accordingly. It should be noted that this was not the aim of the work and, as with their paper Ody et al. 2023, a separate analysis was carried out in each case. Nevertheless, we would happy to provide you with our data after the 15-year retention period if you are interested in a more in-depth analysis of the readiness potential.

Original version:

In controls, the RP in the “Generate Tone” condition showed an early positive amplitude shift compared with the RP in the “Button Alone” condition. In patients, the “Generate Tone” condition resulted in a small but consistent negative potential shift in the time preceding the motor action

Revised version:

In controls, the RP in the “Generate Tone” condition showed a lately negative amplitude shift compared with the RP in the “Button Alone” condition. In patients, the “Button Alone” condition showed a small but consistent negative potential shift in the preceding the motor action compared to the “Generate Tone” condition.

Minor issues:

2# Comment: The results section is a bit confusing. Generally, if the results are reported in the text, they don't need to be reported in a table. I think it would be more effective to report everything in the tables, including both types of degrees of freedom and the effect sizes and then highlight the significant results in the text, for example, ‘As shown in Table 2, there was a significant difference between X and Y, with higher amplitudes in X (estimated marginal mean = xyz) than Y (EMM = xyz).’ Otherwise, please consider consolidating the results in another way to make them easier to read

2# Answer: The ANOVA tables were adjusted by adding Df2 and we shortened the description in several sections the “Results” section. This was marked as red and crossed out in the revised manuscript.

3# Comment: After reading the RP sections in the methods and results, it wasn't clear which time window was analysed. In the methods (lines 255-256), it suggests multiple analyses were conducted. However, in the results, only one value per analysis is given, so I suspect there must be a single selected time window. If this is the case, could the authors please clarify how this window was chosen?

3# Answer: All 10 segments were used in the ANOVA. To make this easier to understand, the period (- 2,000 ms to 0 ms) was again named (page 7 - 8 line 255 - 258 and it was pointed out that the 10 subdivided ANOVA segments were also used as within-subjects variables. We deliberately did not include the results with regard to the 10 segments in the ANOVA because we did not consider it crucial in the context of this research question and were concerned that this would have made the overview of this table more difficult. We have now included it in the ANOVA (See Table 8.).

Original version:

Furthermore, the RPs from the 10 divided segments were analyzed in a three-way repeated measures ANOVA for group (“CG” vs “SG”) as the between-subjects factor and conditions (“Generate Tone” versus “Button Alone”) and electrodes (F3, Fz, F4, C3, Cz, C4, and FCz) as within-subjects variables.

Revised version:

Furthermore, RPs were analyzed in a three-way repeated measures ANOVA with group (“CG” vs “SG”) as between-subjects factor and conditions (“Generate Tone” versus “Button Alone”), electrodes (F3, Fz, F4, C3, Cz, C4, and FCz) and the 10 segments (-2,000 ms to 0 ms in 200 ms intervals) as within-subject variables.

4# Comment: For the ERP analyses, what is the reason for including electrodes as a factor, rather than taking an average across them? Do the authors have a reason to believe that N1/P2/RP would differ between electrodes? If not, I recommend running the analysis on an average of the selected electrodes, as this would make the results easier to parse.

5# Answer: Initially, we considered averaging the selected electrodes; however, the statistical analysis indicated that electrodes were highly significant in the MANOVA for both N1 and P2, neither influenced by other relevant effects nor involved in any interaction effects with groups or conditions. Therefore, we decided to include the electrodes as a separate factor in the MANOVA.

5# Comment: There is a possible error in the text in lines 408/409, where it lists the included conditions as 'generate tone and 'hear tone'. I understand from the earlier text that this analysis included the 'generate tone' and 'button alone' conditions. However, see my above comment regarding including all three conditions

5# Answer: You are right and it is a mistake. This has been corrected and thank you for pointing it out.

6# Comment: For Fig 3 and 4, it would be better to see the individual data points (see, for example, (Rousselet et al., 2016)). This technique is more effective than a simple error bar at showing the dispersion of the data

6# Answer: The graphics have been revised using Dot Density and have been re-uploaded (see figure 3. and 4.).

7# Comment: The authors might consider including a brief discussion regarding the lack of correlation between sensory suppression and clinical symptoms. This has been a consistent difficulty for the field that some consider to be a key piece of lacking evidence for connecting deficits in the forward model to reduced sensory suppression in patients. (Mathalon & Ford, 2012) gives a detailed discussion of this issue.

7# Answer: Thank you very much for this hint and the papers. Reviewer #2 also asked that I perform and complete an analysis of disease duration and cognitive performance. I have added this in the Discussion combined. Please see page 19, line 575 – 589

The new paragraph reads now:

Moreover, we found no association between ERP measures and cognition tests. It should be noted, however, that Mathalon and Ford (73) have already indicated, despite advances in neuroimgaging and other methods, the challenges of linking sensory suppression to the clinical symptoms of schizophrenia remain. Mathalon and Ford suggest several obstacles that might contribute to this lack of correlation, including small sample sizes, unreliable measures and medication effects. These factors prevent a clear understanding of how specific neurobiological abnormalities relate to the various clinical symptoms of schizophrenia. In the discussion of the relationship between illness duration and cognitive performance, a negative correlation was also found between illness duration and the subtest 'Semantic Word Fluency'. In addition, a negative correlation between illness duration and cognition was also observed that just failed significance. It is important to note that the present study exclusively included patients within the age range of 18 to 45 years, and the analysis was further refined by considering the small patient sample for such studies. Nevertheless, other studies have shown that illness Duration is associated with progressive decline in both cognitive and cerebellar function, highlighting the role of disease progression in causing further neurocognitive impairment (74, 75).

8# Comment: Line 127 says the study (Roach, 2023) was ‘aforementioned’, but I think it wasn’t mentioned before this sentence.

8# Answer: The “aformentioned study” refers to the study by Judith Ford et al. 2014, where one of her co-authors made the data from the 2014 study available online. The text passage has been revised to make this clearer. See page 2, line 126 – 128.

Original version:

While the mechanisms of EC/CD and their disruption in schizophrenia have been extensively studied, papers have employed data from the aforementioned study(33), particularly in deep learning (34).

Revised version:

While the mechanisms of EC/CD and their disruption in schizophrenia have been extensively studied, further publications followed, particularly in the area of deep learning (34), with the publication of the data from the Ford et al. 2014 study by one of the co-authors (33).

9# Comment: In line 246, there is an extra quotation mark before “Button Alone”.

9# Answer: Thank you for pointing this out and has been adjusted accordingly.

#Reviewer 2:

Major revision:

1 # Comment: The patient population in this study was diverse, with 14 diagnoses of paranoid schizophrenia, seven of undifferentiated schizophrenia, three of residual schizophrenia, three of schizoaffective disorder and one of hebephrenic schizophrenia. The mean duration of illness is given as 7.74 years, but the range is unclear and should be clarified. In other words, the study includes a diverse group of patients with different backgrounds. It is assumed that cerebellar and cognitive decline progresses with disease duration. The relationship between disease duration and cognitive function should be discussed.

1# Answer: A calculation was made between Illness Duration and cognitive performance. Overall, there was a significant negative correlation between one subtest (“Semantic word fluency”) and Illness duration. Others were non-significant, with “total number of points achieved” (r = - .329, p = .09) and “total number of all failed individual tests” (r = 0.341, p = .08) revised just failing significance. The text at this paragraph was revised according to your and reviewer 1 suggestions. (See page 17, line 575 - 589). The standard deviation of Illness Duration has been added to the table.

The new paragraph reads now:

Moreover, we found no association between ERP measures and cognition tests. It should be noted, however, that Mathalon and Ford (73) have already indicated, despite advances in neuroimgaging and other methods, the challenges of linking sensory suppression to the clinical symptoms of schizophrenia remain. Mathalon and Ford suggest several obstacles that might contribute to this lack of correlation, including small sample sizes, unreliable measures and medication effects. These factors prevent a clear understanding of how specific neurobiological abnormalities relate to the various clinical symptoms of schizophrenia. In the discussion of the relationship between illness duration and cognitive performance, a negative correlation was also found between illness duration and the subtest 'Semantic Word Fluency'. In addition, a negative correlation between illness duration and cognition was also observed that just failed significance. It is important to note that the present study exclusively included patients within the age range of 18 to 45 years, and the analysis was further refined by considering the small patient sample for such studies. Nevertheless, other studies have shown that illness Duration is associated with progressive decline in both cognitive and cerebellar function, highlighting the role of disease progression in causing further neurocognitive impairment (74, 75).

2 # Comment: As the author states in the first paragraph of the Discussion, it has been suggested that changes in CD are associated with positive symptoms in particular. However, the paper does not address the association between positive symptoms on the PANSS and ERPs, which may be representative of changes in CD. If the results did not demonstrate an association, this would still need to be addressed in the discussion. Related to the above point #1, it may be that schizophrenia is thought to be preceded by positive or negative symptoms, and that cognitive decline becomes more pronounced as the illness progresses, but these points also need to be considered

2# Answer: This is a key point which reviewer #1 asked us to revise. This subject is also addressed on pages 17, lines 575 - 589.

Minor revisions:

3# Comment: Title of Table 5 and 6; not “N1” but “P2”.

3# Answer: Thank you for pointing out this error and we have corrected this.

4 # Comment: P13, L369; Does ERP component ("N1 and P2") mean peak-to-peak amplitude? I did not understand this procedure, although I am not familiar with this description.

4# Answer: Exactly, the ERP component („N1 and P2“) means peak to peak amplitude. The relevant passage has been adapted to make it easier to understand. (See Page 13, line 369).

Original version:

To evaluate the specificity of N1 and P2 results, we performed a rep

---

## [Decision Letter · Decision Letter 1]

1 Sep 2025

PONE-D-24-58770R1Corollary discharge and efference copy mechanisms in schizophrenia and controls: The N1 and P2 evoked potential components differentially react to self-initiated tones in schizophreniaPLOS ONE

Dear Dr. Liermann-Koch,

Thank you for submitting your manuscript to PLOS ONE. After careful consideration, we feel that it has merit but does not fully meet PLOS ONE’s publication criteria as it currently stands. Therefore, we invite you to submit a revised version of the manuscript that addresses the points raised during the review process.

We look forward to receiving your revised manuscript.

Kind regards,

Jie Wang, Ph.D.

Academic Editor

PLOS ONE

**Journal Requirements:**

**Additional Editor Comments:**

The authors need to address the comments raised by reviewers.

Reviewers' comments:

Reviewer's Responses to Questions

**Comments to the Author**

1. If the authors have adequately addressed your comments raised in a previous round of review and you feel that this manuscript is now acceptable for publication, you may indicate that here to bypass the “Comments to the Author” section, enter your conflict of interest statement in the “Confidential to Editor” section, and submit your "Accept" recommendation.

Reviewer #1: (No Response)

Reviewer #2: All comments have been addressed

2. Is the manuscript technically sound, and do the data support the conclusions?

Reviewer #1: Yes

Reviewer #2: Yes

3. Has the statistical analysis been performed appropriately and rigorously? 

Reviewer #1: Yes

Reviewer #2: Yes

4. Have the authors made all data underlying the findings in their manuscript fully available?

Reviewer #1: No

Reviewer #2: Yes

5. Is the manuscript presented in an intelligible fashion and written in standard English?

Reviewer #1: Yes

Reviewer #2: Yes

6. Review Comments to the Author

**Reviewer #1: ** Thanks to the authors for addressing my comments. They took them seriously, made appropriate adjustments and the manuscript looks in great shape. I just have a couple of follow up comments.

1. The statistics for the new analysis that I requested (F(1.92,

105.83) = 2.19, p = 0.0116) does seem to show a significant group difference for Readiness Potential but you wrote in the text that there is no significant difference. Could you please clarify? If there indeed was a group difference for RP and not for LRP, that’s a really interesting result which should be addressed briefly in the discussion.

2. Please replace ‘Condition 1, Condition 2’ etc. in Figure 7 with the actual names of the conditions.

**Reviewer #2:**  The manuscript has been well revised. I believe it will be acceptable once a few corrections have been made.

Minor points.

P3, L77 “Schizophrenia is a neuropsychiatric disorder with an estimated incidence rate of approximately 0.01% with onsets,”

Specify the duration of the incidence. I suggest, “Schizophrenia is a neuropsychiatric disorder with an estimated annual incidence rate of approximately 0.01% with onsets,”

7. PLOS authors have the option to publish the peer review history of their article (what does this mean? ). If published, this will include your full peer review and any attached files.

**Do you want your identity to be public for this peer review?** For information about this choice, including consent withdrawal, please see our Privacy Policy .

Reviewer #1: No

Reviewer #2: No

---

## [Author Response · Author response to Decision Letter 2]

14 Oct 2025

Dear colleagues from PLOSone, Dr. Ody and dear Prof. Kazuyori Yagyu,

Thank you very much for your positive response to my expanded analysis, including text passages. I would also like to thank you for reviewing my text with such care and attention, so that the manuscript can be published as a paper. As in the last review, I have created a Word file called “Questions and Answers 2nd Part” in which I respond to the comments made by the two reviewers.

In addition, I was able to make the data available online in collaboration with my university. The data includes the raw data from the EEG examination, preparatory EEG analysis for N1/P2, as well as readiness potential and clinical data. Further information regarding data processing is stored in a README file. The university has also already created a DOI number (https://doi.org/10.17185/duepublico/84480). It should be noted, however, that the university needs approximately four weeks to make the data available online.

Nevertheless, the university has already generated a link so that the data can be reviewed. The hyperlink is https://nxcl.rds.uni-due.de/s/cprFxrGT7K7Pqms (password: review). In the following sections, I will provide feedback on each comment made by the two reviewers. If you have any questions regarding the provision of data, please let us know.

Yours sincerely,

Constantin Liermann-Koch

#Reviewer 1:

Minor issues

1# Comment: Thanks to the authors for addressing my comments. They took them seriously, made appropriate adjustments and the manuscript looks in great shape. I just have a couple of follow up comments.

The statistics for the new analysis that I requested (F(1.92, 105.83) = 2.19, p = 0.0116) does seem to show a significant group difference for Readiness Potential but you wrote in the text that there is no significant difference. Could you please clarify? If there indeed was a group difference for RP and not for LRP, that’s a really interesting result which should be addressed briefly in the discussion.

1# Answer: Thank you very much for bringing this to my attention. I have rechecked the statistical analysis and found that I did not enter the results correctly. Primarily, one decimal place too many was added and a 6 was entered instead of a 9. In addition, it was the analysis for interaction between condition and group. The correct data has now been entered. For verification purposes, I have attached the ANOVA with the 3 conditions for RP and LRP as a separated data (“ANOVA with the 3 conditions for RP and LRP”).

2# Comment: Please replace ‘Condition 1, Condition 2’ etc. in Figure 7 with the actual names of the conditions.

2# Answer: Thank you very much for pointing this out. The graphic in Figure 7 has been adjusted accordingly and re-uploaded.

#Reviewer 2:

Minor revisions:

1# Comment: P3, L77 “Schizophrenia is a neuropsychiatric disorder with an estimated incidence rate of approximately 0.01% with onsets,” Specify the duration of the incidence. I suggest, “Schizophrenia is a neuropsychiatric disorder with an estimated annual incidence rate of approximately 0.01% with onsets,”

1# Answer: Thank you very much for suggesting a corrective to this sentence, so that this statement can be properly classified.

---

## [Decision Letter · Decision Letter 2]

20 Oct 2025

Corollary discharge and efference copy mechanisms in schizophrenia and controls: The N1 and P2 evoked potential components differentially react to self-initiated tones in schizophrenia

PONE-D-24-58770R2

Dear Dr. Liermann-Koch,

We’re pleased to inform you that your manuscript has been judged scientifically suitable for publication and will be formally accepted for publication once it meets all outstanding technical requirements.

Kind regards,

Jie Wang, Ph.D.

Academic Editor

PLOS ONE

Additional Editor Comments (optional):

Reviewers' comments:

Reviewer's Responses to Questions

**Comments to the Author**

1. If the authors have adequately addressed your comments raised in a previous round of review and you feel that this manuscript is now acceptable for publication, you may indicate that here to bypass the “Comments to the Author” section, enter your conflict of interest statement in the “Confidential to Editor” section, and submit your "Accept" recommendation.

Reviewer #1: All comments have been addressed

2. Is the manuscript technically sound, and do the data support the conclusions?

Reviewer #1: Yes

3. Has the statistical analysis been performed appropriately and rigorously? 

Reviewer #1: Yes

4. Have the authors made all data underlying the findings in their manuscript fully available?

Reviewer #1: Yes

5. Is the manuscript presented in an intelligible fashion and written in standard English?

Reviewer #1: Yes

6. Review Comments to the Author

Reviewer #1: The authors addressed all of my concerns and I'm very happy to recommend that this paper is accepted.

7. PLOS authors have the option to publish the peer review history of their article (what does this mean? ). If published, this will include your full peer review and any attached files.

**Do you want your identity to be public for this peer review?** For information about this choice, including consent withdrawal, please see our Privacy Policy .

Reviewer #1: No

---

## [Editor Report · Acceptance letter]

PONE-D-24-58770R2

PLOS ONE

Dear Dr. Liermann-Koch,

I'm pleased to inform you that your manuscript has been deemed suitable for publication in PLOS ONE. Congratulations! Your manuscript is now being handed over to our production team.

Kind regards,

on behalf of

Dr. Jie Wang

Academic Editor

PLOS ONE